# Monitoring Marine Oil Spills in Hyperspectral and Multispectral Remote Sensing Data by the Spectral Gene Extraction (SGE) Method

**Dong Zhao [1], Bin Tan [2,3], Haitao Zhang [4] and Rui Deng [1,5,*]**

[1] Key Laboratory of Exploration Technologies for Oil and Gas Resources, Ministry of Education, Yangtze University, Wuhan 430100, China

[2] Key Laboratory of Natural Resources Monitoring in Tropical and Subtropical Area of South China, Ministry of Natural Resources, Zhuhai 519082, China

[3] Surveying and Mapping Institute Lands and Resource Department of Guangdong Province, Guangzhou 510500, China

[4] College of Resources and Environment, Henan University of Economics and Law, Zhengzhou 450011, China

[5] State Key Laboratory of Oil and Gas Reservoir Geology and Exploitation, Chengdu University of Technology, Chengdu 610059, China

* Correspondence: dengrui@yangtzeu.edu.cn

**Abstract:** Oil spill incidents threaten the marine ecological environment. Detecting sea surface oil slicks by remote sensing images provides support for the efficient treatment of oil spills. This is important for sustainable marine development. However, traditional methods based on field analysis are time-consuming. Spectral indices lack applicability. In addition, traditional machine learning methods strictly rely on training and testing samples which are in short supply in oil spill images. Inspired by the spectral DNA encoding method, a spectral gene extraction (SGE) method was proposed to detect oil spills in hyperspectral images (HSI) and multispectral images (MSI). The SGE method contained a parameter and two strategies. The parameter of elimination was designed based on the population genetic frequency. It was used to control the number of spectral genes. The spectral gene extraction strategies, named largest in-class similarity (LIS) strategy and largest inter-class difference (LID) strategy, were proposed to mine the spectral genes by oil spill samples. The oil spills would be determined by calculating the similarity of the extracted spectral genes to the DNA encoded images. In this research, the SGE method was validated by two AVIRIS images of the Gulf of Mexico oil spill, one MODIS image of the Gulf of Mexico oil spill, and one Landsat 8 image of a Persian Gulf oil spill. The oil spills in different remote sensing images could be detected accurately by the proposed method in a small set of samples. Experimental results indicated that the proposed method was suitable for detecting marine oil spills in AVIRIS, MODIS, and Landsat 8 images. In addition, the SGE method with the LIS strategy was more suitable for detecting oil spills in HSI. Its proper elimination rates were 0.8~1.0. The SGE method with the LID strategy was more suitable for detecting oil spills in MSI. Its proper elimination rates were 0.5~0.7.

**Keywords:** hyperspectral remote sensing; multispectral remote sensing; oil spill detection; spectral gene extraction





## 1. Introduction

Oil spill incidents occur regularly. In 2010, the oil drilling platform of Deepwater Horizontal (DWH) exploded and about 3.2 million barrels of crude oil leaked into the Gulf of Mexico. At least 2500 km$^2$ of seawater were covered in oil and a large amount of marine life died because of the oil spill. In 2020, a large Japanese cargo ship named WAKASHIO struck a reef off Mauritius and leaked a large amount of fuel oil. This oil spill incident caused the most serious environmental disaster for Mauritius. Oil spill incidents usually cause serious marine ecological disasters. They make crude oil or fuel oil leak into the

marine environment. According to recent studies by Alloy and Xu [1,2], the composition of crude oil will seriously harm marine life, especially fertilized eggs and biological larvae. Finally, toxic substances will be deposited in the human body [3–5]. Monitoring oil spills on the sea surface rapidly contributes to measuring pollution emergencies caused by oil spill incidents. Multispectral remote sensing technology is an effective approach to detect oil spills on a large scale [6]. Airborne hyperspectral spectrometers can obtain near-continuous spectral signals and detect different thicknesses of oil slicks [7]. Airborne hyperspectral technology can also reduce the restrictions of current weather. Thus, it is important for the sustainable development of the marine environment to monitor oil spills in hyperspectral images (HSI) and multispectral images (MSI).

In 2010, Clark et al. mapped the oil spill caused by the incident of DWH in airborne visible infrared imaging spectrometer (AVIRIS) images by the near-infrared absorption spectral features of hydrocarbons based on field analyses [8]. Methods based on field analyses were usually time-consuming because they needed plenty of time to find and evaluate the spectral features artificially [9]. In 2012, Loos et al. proposed a fluorescence index (FI) and a rotation absorption index (RAI) to distinguish between oil spills and false alarms in MODIS images [10]. In 2014, Zhao et al. proposed a floating algae index (FAI) to detect oil spills on the sea surface in MODIS, MERIS, and Landsat data [11]. These detection methods based on spectral indices favored the detection of thicker oil slicks, such as emulsions and continuous true color oil slicks (Code 5). In addition, spectral indices were inapplicable for different datasets. In 2019, Liu compared the oil spill detection ability of backpropagation neural networks (BP-NN), support vector machines (SVM), and stacked auto-encoders (SAE) [12]. He stated that the SAE method combined with spatial information obtained better oil spill detection performance than SVM and BP-NN. However, the traditional machine learning methods needed many samples to train the model and then produce oil spill detection results. At present, the urgent task was constructing a rapid oil spill detection method that was applicable to different datasets. In addition, the detection method should be able to produce satisfying results in a small set of samples since the oil spill areas were generally limited in MSI.

Inspired by pattern recognition technologies and spectral encoding methods [13,14], a spectral gene extraction (SGE) method based on the spectral DNA encoding method was proposed for detecting oil spills in this research. It was validated that the spectral DNA encoding method could describe spectral information distinctively [15,16]. Based on this, the SGE method was proposed to mine the population genetic characteristics of oil slicks. First, the remote sensing images and the training samples were encoded by the spectral DNA encoding method. The continuous spectral information would be translated into discrete code words. Then, a largest in-class similarity (LIS) strategy and a largest inter-class difference (LID) strategy were proposed to extract the spectral genes in the spectral DNA chains of training samples. The extracted spectral genes were the spectral details of each category without redundant spectral DNA information. In this step, a parameter of the elimination rate was used to control the spectral gene selection process. Next, the similarity between the spectral genes of samples and the DNA-encoded pixels of the HSI would be calculated. Then, the category of the pixels would be determined according to the similarity calculation results. Finally, the oil spill detection results would be produced based on the results calculated by the LIS and LID strategies. The innovation of the proposed method was that it could rapidly mine the spectral differential details of oil spills for the recognition process. It was more stable than the spectral indices. In addition, it was suitable for detecting oil spills in different data by a few samples.

In this research, the proposed method was validated through two AVIRIS images obtained during the Gulf of Mexico oil spill incident in 2010, one MODIS image of the Gulf of Mexico oil spill incident in 2010, and one Landsat 8 image of a Persian Gulf oil spill in 2013. The experimental results showed that the SGE method could correctly detect the oil spills in different HSI and MSI. The description of the proposed method, experimental results, discussions, and conclusions are provided below.

## 2. Materials and Methods

### 2.1. Oil Spills

According to the Bonn Agreement Oil Appearance Code [17], the thicknesses of oil slicks formed by oil spill incidents are classified as silver (Code 1, <0.3 microns), rainbow (Code 2, 0.3~5.0 microns), metallic (Code 3, 5.0~50 microns), discontinuous true oil color oil slicks (Code 4, 50~200 microns), continuous true oil color oil slicks (Code 5, >200 microns), and emulsions (Figure 1). Since the thickness of an emulsion is unstable and unpredictable, emulsions are not coded. It is difficult to observe silver, rainbow, and metallic respectively in HSI or MSI. Thus, oil slicks of codes 1–3 are usually divided into sheens during the detection procedure [18].

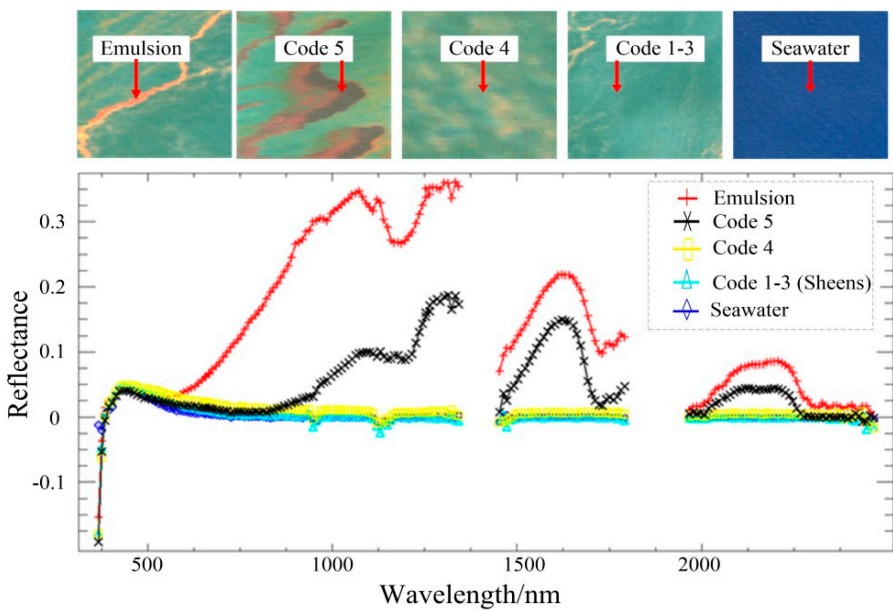

**Figure 1.** Spectral curves of seawater and different thicknesses of oil slicks in an AVIRIS image.

Different thicknesses of oil slicks have different visual, shape, and spectral characteristics. Sheens alter the color of the seawater slightly. Thus, the spectra of sheens are very similar to that of seawater. Because of the diffusion and the viscosity of oil slicks, sheens usually surround thick oil slicks on a large scale. Oil slicks of Code 4 look yellow or orange. They appear patchy in the true color composite images. Their infrared reflectance is slightly higher than that of seawater and sheens [19–21]. Oil slicks of Code 5 cannot exist in large quantities on the sea surface due to their weight [22]. They are black and show distinct infrared spectral reflectance compared to seawater, sheens, and oil slicks of Code 4 [23]. Emulsions are weathered oil slicks [24]. They look orange or red and are usually distributed in long convergence zones. Sometimes, the convergence zones of emulsions extend for several kilometers continuously [25]. Emulsions exhibit much higher infrared spectral reflectance than the other oil slicks [26–29]. It should be noticed that sun glint is very important for oil spill detection. Too much or too weak sun glint will make the oil slicks unrecognizable [30–32].

### 2.2. Spectral DNA Encoding Method

The genetic information of biota is stored in DNA. The compositions of DNA include four types of bases: thymine (T), adenine (A), cytosine (C), and guanine (G) [33]. Different arrangements of the bases represent different genetic information which creates biological diversity. Inspired by this, the spectral DNA encoding method was studied for object recognition in HSI for many years [34,35].

The spectral DNA encoding process includes two aspects [15]: spectral brightness encoding and spectral shape encoding, according to Hongzan Jiao. During the brightness

encoding process, the average spectral reflectance values are calculated by Formula (1). Then, another two gradients are calculated on the basis of the averages of Formulas (2)–(5). Finally, the continuous-valued reflectance of spectra will be converted into DNA code words according to the calculated gradients by Formula (6).

$$T_{middle} = \rho \sum_{i=1}^{Nb} \frac{y_i}{Nb} \tag{1}$$

$$T_{higher} = \sum_{i=1}^{Nb} \frac{Bigger(y_i)}{K} \tag{2}$$

$$Bigger(y_i) = \begin{cases} y_i, \text{ if } y_i \geq T_{middle} \\ 0, \text{ else} \end{cases} \tag{3}$$

$$T_{lower} = \sum_{i=1}^{Nb} \frac{Smaller(y_i)}{P} \tag{4}$$

$$Smaller(y_i) = \begin{cases} y_i, \text{ if } y_i \prec T_{middle} \\ 0, \text{ else} \end{cases} \tag{5}$$

$$DNA_i^{brightness} = \begin{cases} G, \text{ if } y_i \in [T_{higher}, y_{\max}] \\ A, \text{ if } y_i \in [T_{middle}, T_{higher}) \\ C, \text{ if } y_i \in [T_{lower}, T_{middle}) \\ T, \text{ if } y_i \in [y_{\min}, T_{lower}) \end{cases} \tag{6}$$

In the formulas, Nb is the spectral band number of samples and pixels. $y_i$ is the reflectance of the i-th band, $i \in [1, Nb]$. In Formula (1), $\rho$ is the adaptive coefficient of the DNA brightness encoding process. It is used to make the encoding process more applicable. In Formula (2), K is the number of bands whose reflectance is bigger than $T_{middle}$. In Formula (4), P is the number of bands whose reflectance is smaller than $T_{middle}$. In Formula (6), $y_{\min}$ and $y_{\max}$ are the minimum and the maximum values of the spectral reflectance, respectively. After encoding, the spectral brightness DNA strands are built up through the encoded DNA code words.

During the shape encoding process, the tolerance value ($\Delta$) of the spectra is calculated by Formula (7). Then, the descriptions of spectral shapes are defined according to $\Delta$. The definitions proposed by Chang Chein-I are adopted in this research [35]. Here are the types of spectral shape patterns in this research [15]:

Type 1 : $|y_i - y_{i-1}| \leq \Delta$ and $|y_{i+1} - y_i| \leq \Delta$
Type 2 : $|y_i - y_{i-1}| \leq \Delta$ and $|y_{i+1} - y_i| > \Delta$ or $|y_i - y_{i-1}| > \Delta$ and $|y_{i+1} - y_i| \leq \Delta$
Type 3 : $y_i - y_{i-1} < -\Delta$ and $y_{i+1} - y_i < -\Delta$ or $y_i - y_{i-1} > \Delta$ and $y_{i+1} - y_i > \Delta$
Type 4 : $y_i - y_{i-1} < -\Delta$ and $y_{i+1} - y_i > \Delta$ or $y_i - y_{i-1} > \Delta$ and $y_{i+1} - y_i < -\Delta$

Finally, the spectral shape information will be converted to DNA code words according to Formula (8).

$$\Delta = \theta \left( \frac{1}{Nb - 1} \right) \sum_{i=2}^{Nb} (y_i - y_{i-1}) \tag{7}$$

$$DNA_j^{shape} = \begin{cases} G, \text{ if } y_j \in Type4 \\ A, \text{ if } y_j \in Type3 \\ C, \text{ if } y_j \in Type2 \\ T, \text{ if } y_j \in Type1 \end{cases} \tag{8}$$

$$DNA^{chain} = \left\{ DNA_i^{brightness}, DNA_j^{shape} \right\} \tag{9}$$

In Formula (7), $\theta$ represents the shape adjacent coefficient. $\theta$ makes the encoding process adjust the changes of spectral shape. The number of spectral shape code words is

(Nb-2) because three adjacent bands determine one spectral shape code. After encoding, spectral shape DNA strands would be built up according to the spectral shape code words.

After the spectral brightness and shape information is encoded, it will be connected to constitute a complete spectral DNA chain according to Formula (9). In Formula (9), i ∈ [1, Nb] and j ∈ [1, Nb-2]. Figure 2 intuitively exhibits the spectra of oil slicks and their spectral DNA chains. The first half of the spectral DNA chain is the spectral brightness information, and the second half of the spectral DNA chain is the spectral shape information. In the first half of Figure 2B–F, it can be clearly observed that the spectral DNA chains reflect the peaks of the spectra. However, the second half of Figure 2B–F cannot reflect the spectral shapes directly because the encoded spectral shape DNA code words are determined by three adjacent bands. Thus, the spectral shape DNA information is more intricate.

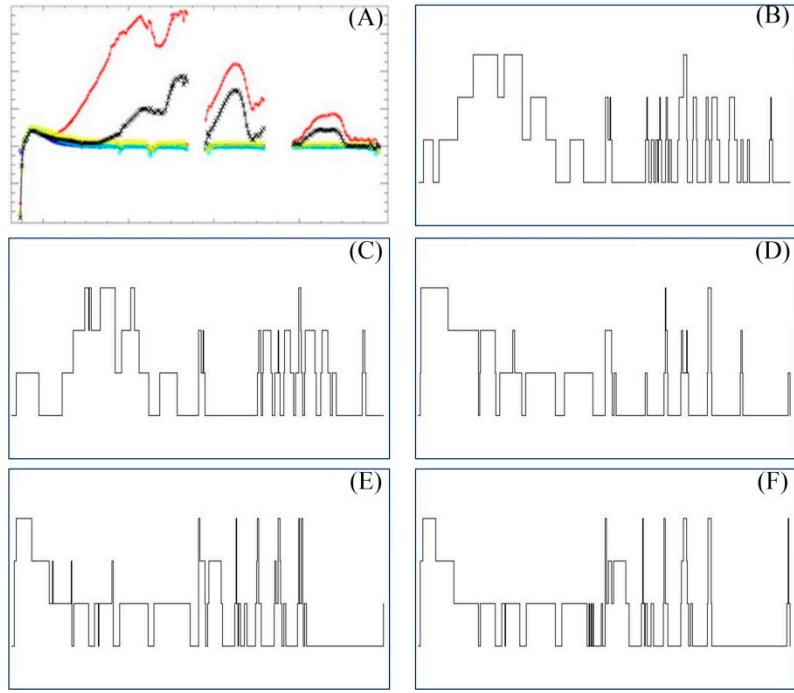

**Figure 2.** The original spectral DNA chains of oil slicks in the experimental AVIRIS image. (**A**) The spectra of oil slicks and seawater, (**B**) the spectral DNA of emulsions, (**C**) the spectral DNA of oil slick of code 5, (**D**) the spectral DNA of oil slick of code 4, (**E**) the spectral DNA of sheens, and (**F**) the spectral DNA of seawater.

### 2.3. SGE Method

Detecting oil spills by the original spectral DNA chains performs poorly because they contain much redundant information which dilutes the identification ability. Thus, this research proposed the SGE method with two strategies (LIS strategy and LID strategy) to extract the differential spectral DNA code words in the DNA chains. The extracted differential spectral DNA code words were called spectral genes because they could express the category-specific information like that carried by genes in DNA.

Operator & was designed to obtain the spectral code words shared by the spectral DNA chains of each category, and operator ⊕ was designed to obtain the unique spectral code words of the spectral DNA chains of each category. Figure 3 simulates the calculation process of the proposed operators. The spectral non-genetic strands were represented as blank in Figure 3.

$$DNA_k^{Chain,a} \& DNA_k^{Chain,b} = \begin{cases} DNA_k^{Chain,a}, & if \ DNA_k^{Chain,a} = DNA_k^{Chain,b} \\ 0, & if \ DNA_k^{Chain,a} \neq DNA_k^{Chain,b} \end{cases} \quad (10)$$

$$DNA_k^{Chain,a} \oplus DNA_k^{Chain,b} = \begin{cases} 0, \; if \; DNA_k^{Chain,a} = DNA_k^{Chain,b} \\ DNA_k^{Chain,a}, \; if \; DNA_k^{Chain,a} \neq DNA_k^{Chain,b} \end{cases} \quad (11)$$

$$G_k^{oil} = \begin{cases} Sgn\left[F(DNA_k^{Chain,a}) - e\right], LIS \; strategy \\ Sgn\left[1 - e - F'(DNA_k^{Chain,a})\right], \; LID \; strategy \end{cases} \quad (12)$$

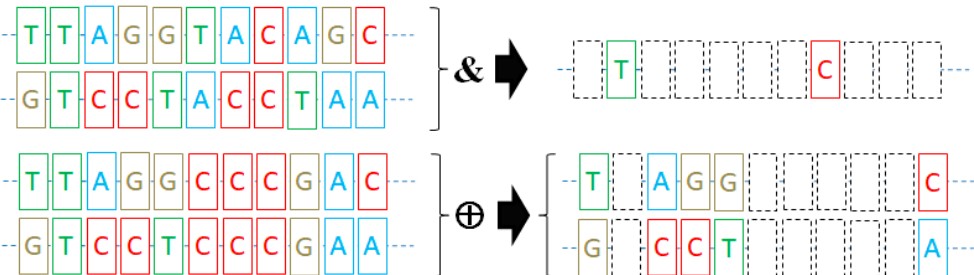

**Figure 3.** Spectral gene extraction strategy schematic diagram.

In Formulas (10)–(12), a and b represent different spectral DNA chains, k was the index of the spectral DNA code words in the spectral DNA chains, and '0' was the non-genetic code word. In Formula (10), if the code words of the k-th band in the spectral DNA chains a and b were the same, the calculation result would be the shared code word. Otherwise, the calculation result would be '0'. In Formula (11), if the code words of the k-th band in the spectral DNA chains a and b were the same, the calculation result would be '0'. Otherwise, the calculation result would be the first specific code word. When the samples of one category were calculated with the operator &, only the spectral DNA code words shared by all the samples would be reserved. When the samples of different categories were calculated with the operator ⊕, only the unique spectral DNA code words of each category would be reserved. In Formula (12), e was the elimination rate. For the LIS strategy, if the frequencies of the reserved code words in the samples of one category population were bigger than e, the reserved code words would be extracted as spectral genes. Otherwise, they would be regarded as non-genetic code words. For the LID strategy, if the frequency of the reserved code words in the samples of the other category populations was smaller than (1 − e), the reserved code words would be extracted as spectral genes. Otherwise, they would be regarded as non-genetic code words.

$$S_{a,b} = \sum f(DNA_k^{Chain,a}, DNA_k^{Chain,b}) \quad (13)$$

$$f\left(DNA_k^{Chain,a}, DNA_k^{Chain,b}\right) = \begin{cases} 1, \; DNA_k^{Chain,a} = DNA_k^{Chain,b} \\ 0, \; DNA_k^{Chain,a} \neq DNA_k^{Chain,b} \end{cases} \quad (14)$$

Formulas (13) and (14) were constructed to evaluate the similarity of the spectral DNA chains and the spectral genes. In the formulas, $f$ was used to calculate the difference of spectral code words. If the code words in the same index were the same, the calculation result would be 1. Otherwise, the result would be 0. The calculation result of '0' (non-genetic code word) and any other spectral DNA code words was 0. The value of $S$ indicated the similarity of the spectral DNA chains and the spectral genes. It could be used to match the sample spectral genes and the spectral DNA chains of pixels. The process of oil spill detection by the SGE method is exhibited in Figure 4. Step 1: Select training samples of seawater and oil slicks from the experimental images. Then, produce the code words of samples and images by the DNA encoding method. Step 2: Extract the spectral genes of oil slicks by the SGE method. The LIS strategy and LID strategy would be used in this step. In addition, the elimination rate would be used to control the gene extraction frequency. Step 3: If the spectral genes were extracted by the LIS strategy, calculate the similarity of the samples and pixels. If the spectral genes were extracted by the LID strategy, determine

the pixels by the sample genes and decision tree (DT) method. Step 4: Produce oil spill detection results using the calculation results in step 3. If the detection results were not satisfying, the process would go back to step 2 and adjust the elimination rate.

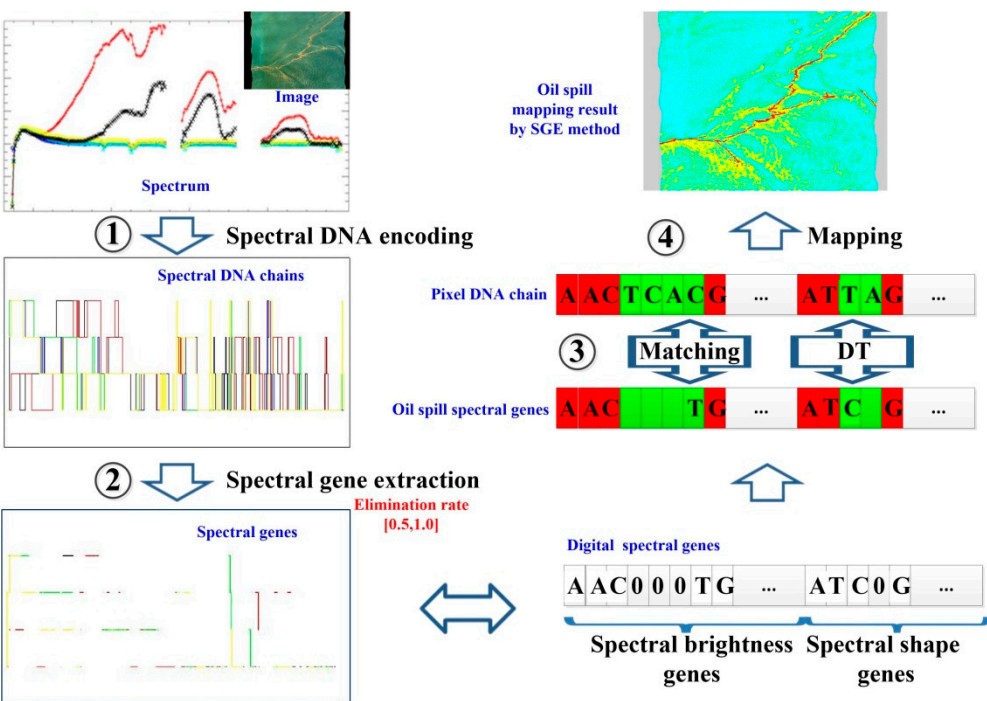

**Figure 4.** The process of oil spill mapping by the SGE method.

## 3. Results

### 3.1. Experimental Data

In April 2010, the DWH oil drilling platform in the Gulf of Mexico (GoM) exploded and a lot of crude oil leaked into the marine environment [19,20]. Two AVIRIS images captured from the GoM oil spill incident were used to evaluate the SGE method. Experimental image 1 was captured on 17 May 2010, UTC 20:12 (solar elevation 56.5 and solar azimuth 261.52). It contained 776 samples and 21,193 lines. Its pixel size was 7.6 m. Experimental image 2 was captured on 6 May 2010, UTC 19:60 (solar elevation 57.52 and solar azimuth 253.81). It contained 783 samples and 26,870 lines. Its pixel size was 7.6 m. Their flight names, on the official website, were f100517t01p00r11 and f100506t01p00r18, respectively. The scopes of the AVIRIS experimental data are displayed in Figure 5. In addition, a MODIS image of the GoM oil spill incident (29 April 2010) and a Landsat 8 image of a Persian Gulf oil spill incident (26 August 2013) were used in this research to validate the applicability of the proposed method.

The experimental images were preprocessed through fast line-of-sight atmospheric analysis of a spectral hypercubes (FLAASH) model in ENVI 5.5. For the correction process, the atmospheric model was tropical, and the aerosol model was marine time. The proposed method was programmed using visual C++. Since the original remote sensing images were too large for the program to handle, subareas were extracted for the experiments.

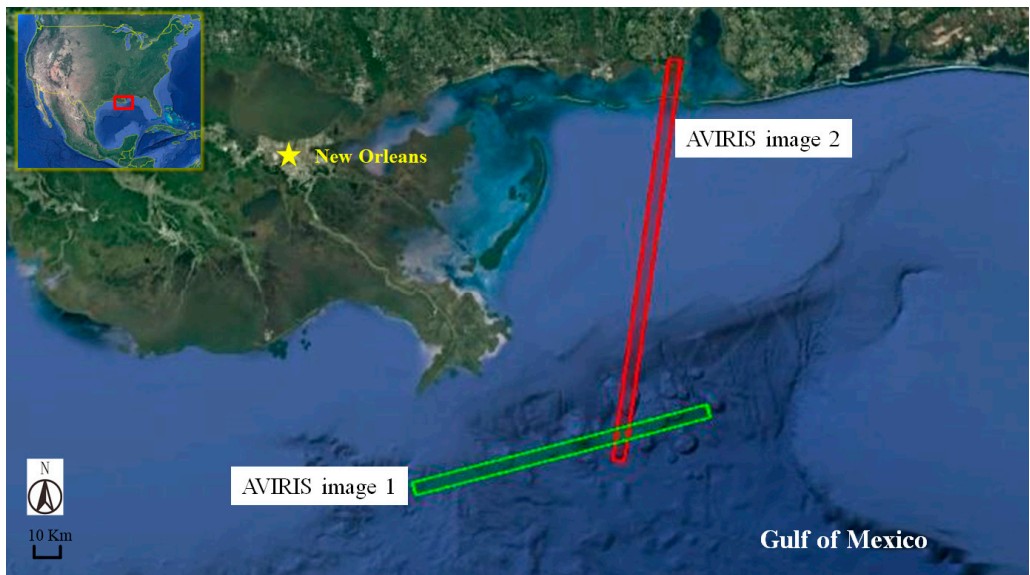

**Figure 5.** The location of the oil spill incident was near the coast of New Orleans. The green boxed area is the scope of experimental image 1 and the red boxed area is the scope of experimental image 2.

*3.2. Experimental Results*

The original AVIRIS experimental image 1 had 19,736 lines and 886 columns (1010.0 km$^2$). The original AVIRIS experimental image 2 contained 26,870 lines and 783 columns (1215.2 km$^2$). Three subareas were extracted from the experimental image 1. They accounted for 12% of the experimental image 1. Since the oil slicks that existed in the experimental image 2 were in a small scale, one subarea was selected from it. The extracted subareas and their oil spill detection results produced by the SGE method with the LIS strategy are shown in Figure 6 (elimination rate = 0.8~1.0). The training samples of the AVIRIS experimental images are listed in Table 1. In the experiments, the training samples were selected from the complete experimental images evenly by points. Only a small number of training samples were collected for the experiments because the thicker oil slicks were distributed in limited areas with blurred borders. Training in large quantities might introduce error samples. The noise spectral DNA strands of error samples would eliminate the pivotal spectral genes, and it would directly reduce the detection accuracy.

**Table 1.** Training samples (pixels) of AVIRIS experimental image 1 and 2.

|  | Emulsion | Code 5 | Code 4 | Code 1–3 | Seawater |
|---|---|---|---|---|---|
| AVIRIS image 1 | 46 | 63 | 26 | 66 | 27 |
| AVIRIS image 2 | 70 | 14 | 51 | 38 | 21 |

In the first subarea (Figure 6A), emulsions were orange and distributed in long stripes. Continuous true color oil slicks (Code 5) were black and distributed in short black strands. Discontinuous true color oil slicks (Code 4) appeared patchy next to emulsions [19,20]. In addition, sheens formed by the diffusion of thicker oil slicks covered the surrounding sea surface [19]. The detection results (Figure 6B) were quite consistent with the oil slicks in the first subarea. Although there were some pixels covered by sheen misidentified as seawater, the majority of sheens were identified correctly. A long convergence of emulsions existed in the second subarea (Figure 6C). Discontinuous true color oil slicks were distributed on both sides of the convergence [23]. In addition, the rest of the second subarea was full of sheens. In the results of this subarea (Figure 6D), the emulsions, discontinuous true color oil slicks, and sheens were all detected accurately. The third subarea (Figure 6E) was the boundary of the oil spill. Only sheens and uncontaminated seawater existed in this subarea. The color of the sea surface covered by sheens was slightly different from that of the unpolluted

seawater [26]. Since the sheens near the boundary were extremely thin, the borderline could not be observed clearly. However, the SGE method extracted abundant spectral genes to differentiate sheens from the unpolluted seawater (Figure 6F). The subareas extracted from the experimental image 1 contained all kinds of oil slicks. They could validate the detection performance of the proposed method effectively.

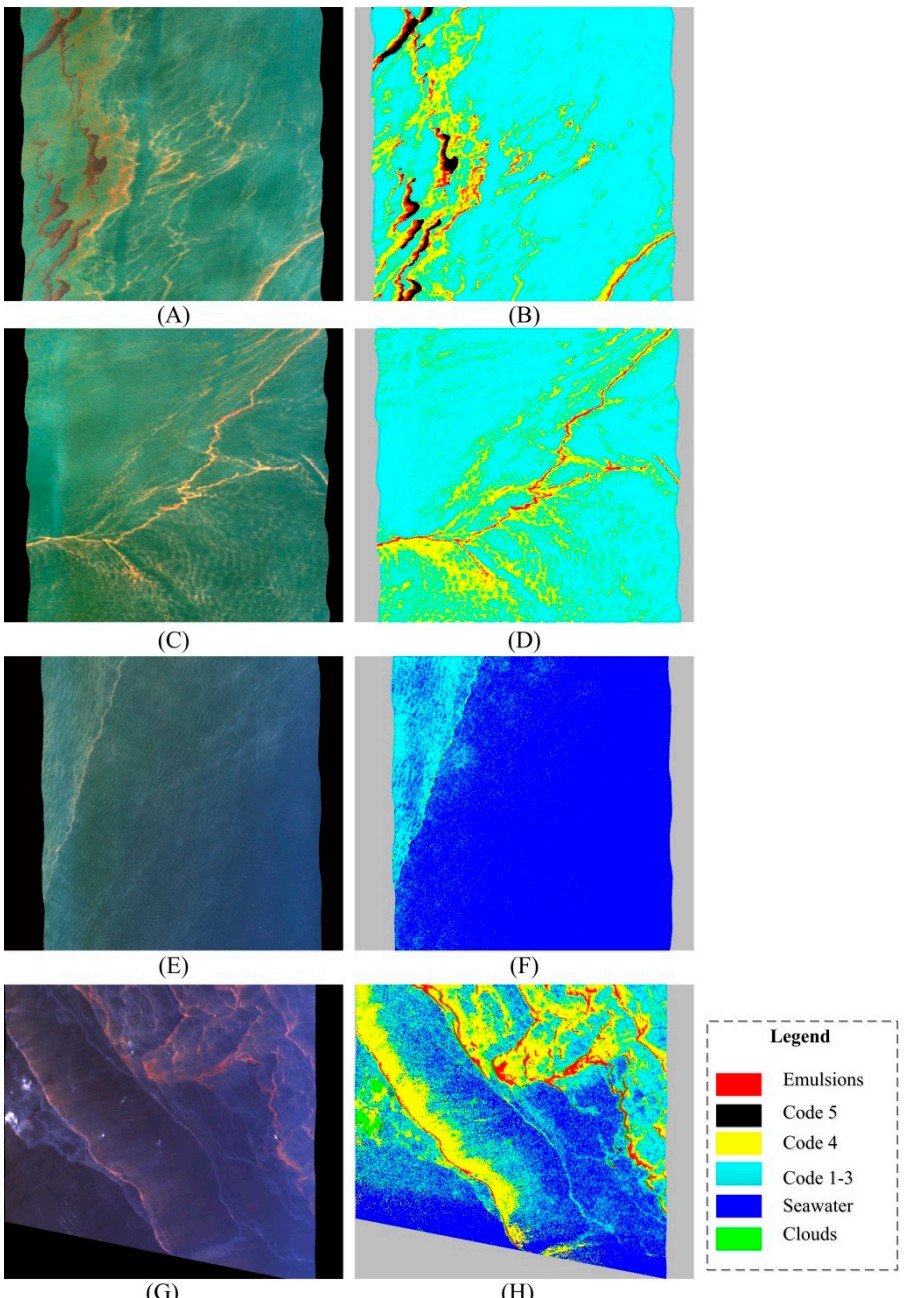

**Figure 6.** The AVIRIS images of the GoM oil spill in 2010 and their detection results by the SGE method. (**A**) Subarea 1 of the AVIRIS experimental image 1; (**B**) Detection result of the subarea 1; (**C**) Subarea 2 of the AVIRIS experimental image 1; (**D**) Detection result of the subarea 2; (**E**) Subarea 3 of the AVIRIS experimental image 1; (**F**) Detection result of the subarea 3; (**G**) Subarea of the AVIRIS experimental image 2; (**H**) Detection result of the subarea of the AVIRIS experimental image 2.

The fourth subarea (Figure 6G) was extracted from the AVIRIS experimental data 2. This subarea contained emulsions, discontinuous true color oil slicks, sheens, seawater, and clouds. Affected by the wind, oil slicks in the subarea 4 diffused on one side and converged

on the other side [25]. Since the AVIRIS experimental data 2 had a different shooting time, shooting angle, weather conditions, etc., from the AVIRIS experimental data 1, the spectra of oil slicks in the two AVIRIS images were quite different (Figures 1 and 7) [24]. However, it could be found from the result of the fourth subarea (Figure 6H) that the stripes of emulsions could be accurately detected. Even though the thicknesses of discontinuous true color oil slicks and sheens were gradually varied, the gradient phenomenon of the oil slicks could be detected clearly by the SGE method.

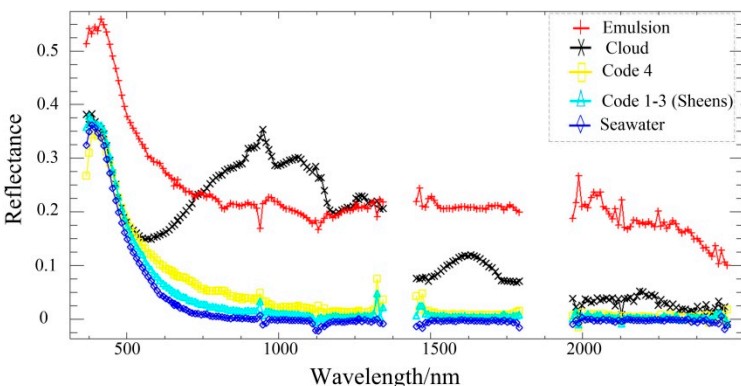

**Figure 7.** Spectra of oil slicks and seawater collected from the experimental AVIRIS image 2.

In this research, MSIs captured by satellites were used to validate the correctness and applicability of the SGE method. Figure 8 shows the GoM oil spill in a MODIS image and its detection result produced by the SGE method with the LID strategy (elimination rate = 0.7). Only 42 samples of oil slicks were used to produce the oil spill detection results. In addition, Figure 9 shows a Persian Gulf oil spill in a Landsat 8 image and its detection result. The result of the Persian Gulf oil spill was produced by the SGE method with LID strategy (elimination rate = 0.5) in 82 samples of oil slicks. Although different thicknesses of oil slicks could not be distinguished in satellite MSI, oil-spill-polluted areas could be identified by the SGE method generally. It indicated that the proposed method was applicable for different HSI and MSI.

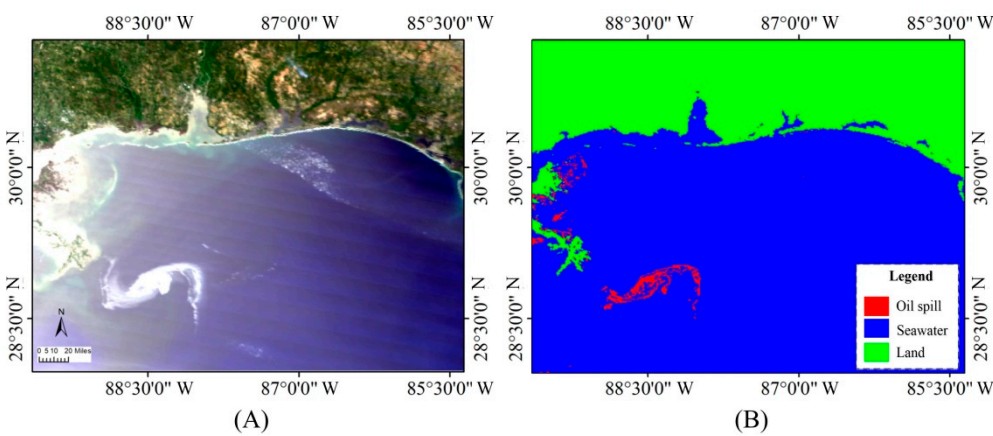

**Figure 8.** MODIS image of the GoM oil spill (29 April 2010) and its detection results produced by the SGE method. (**A**) MODIS image of the GoM oil spill; (**B**) Detection result of the GoM oil spill.

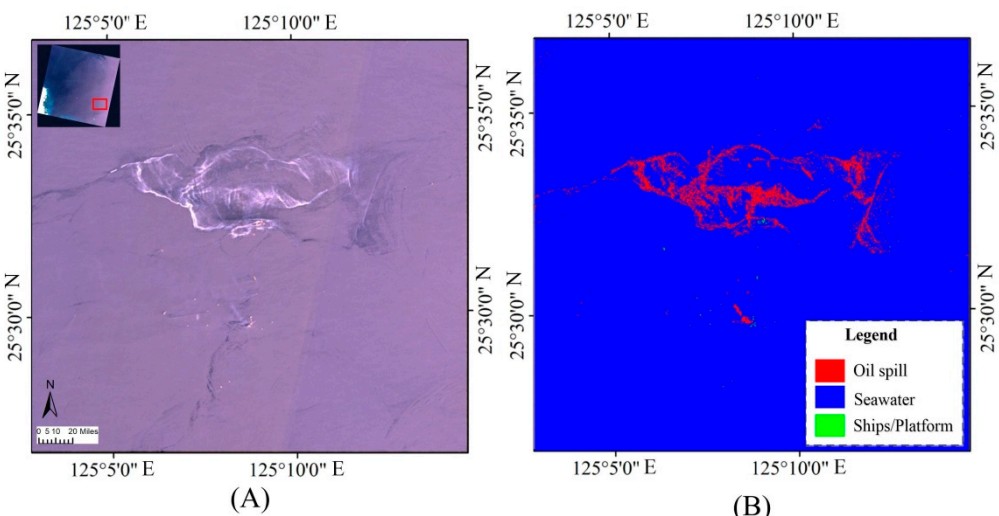

**Figure 9.** Landsat 8 image of a Persian Gulf oil spill (26 August 2013) and its detection results produced by the SGE method. (**A**) Landsat 8 image of a Persian Gulf oil spill; (**B**) Detection result of the Persian Gulf oil spill.

## 4. Discussion

### 4.1. Spectral Gene Extraction Strategies

The SGE method included two spectral gene extraction strategies. The LIS strategy was based on the idea that the spectral DNA strands shared by the populations of one category were the spectral gene information. The LID strategy was based on the idea that the unique spectral DNA strands of the population of one category were the spectral gene information. The pixel of the HSI and MSI would be determined by the similarity between the extracted spectral genes and the pixel DNA code words. Figure 10 shows the oil spill detection results produced by the spectral genes (extracted by the LIS strategy, elimination rate = 1.0) and the original spectral DNA chains. The original spectral DNA chains could not distinguish between the oil slicks with similar spectral signals, such as the sheens and oil slicks of code 4, because the original spectral DNA chains contained a lot of redundant information which diluted the expression ability for the spectra of the oil slick (Figure 2). However, the SGE method could filter the redundant spectral DNA code words by the LIS and LID strategies, and the key spectral genes were reserved to detect oil slicks (Figure 11). The expression ability for oil slicks was strengthened by the spectral genes. Thus, the original spectral DNA chains tended to produce misidentified pixels of oil slicks. It could be clearly observed that the detection result produced by the spectral genes was much more accurate than the result produced by the original spectral DNA chains.

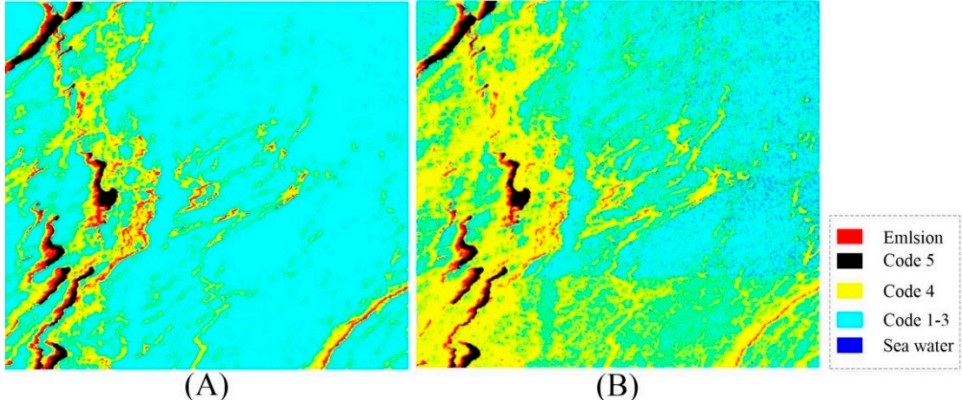

**Figure 10.** Oil spill detection results of the experimental AVIRIS image 1 produced by the spectral genes (**A**) and the original spectral DNA chains (**B**).

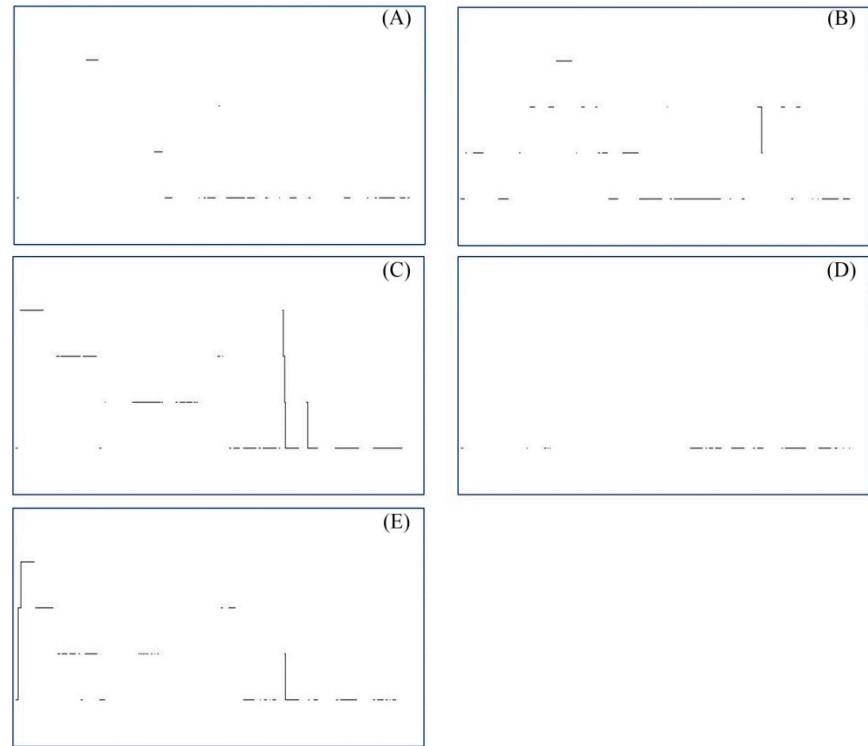

**Figure 11.** The spectral genes extracted by the LIS strategy from the experimental AVIRIS image. (**A**) the spectral genes of emulsions, (**B**) the spectral genes of oil slick of code 5, (**C**) the spectral genes of oil slick of code 4, (**D**) the spectral genes of sheens, and (**E**) the spectral genes of seawater.

However, the spectral genes extracted by the LIS strategy and LID strategy were different. The spectral genes of oil slicks and seawater extracted from the experimental AVIRIS image 1 by the LIS strategy (elimination rate = 1.0) and LID strategy (elimination rate = 1.0) are exhibited in Figures 11 and 12, respectively. In the figures, the spectral genes distributed in pieces and non-genetic strands were blank. It was found from Figures 11 and 12 that the LIS strategy extracted more spectral genes than the LID strategy in HSI. In addition, the spectral genes extracted by the LIS strategy were distributed in both sides of the spectral brightness information part and the spectral shape information part of the spectral DNA chains. However, the LID strategy obtained fewer spectral genes than the LIS strategy, especially for seawater (16) and sheens (14). This might be because the spectral gene extraction criteria of the LID strategy were more demanding than the LIS strategy. In addition, the extracted spectral genes by the LID strategy tended to be inhomogeneous. Thus, the SGE method with the LIS strategy was more suitable for detecting oil slicks in HSI.

Although the spectral genes extracted by the LID strategy were not suitable for detecting oil spills in HSI, they were discovered to be effective for recognizing oil slicks in MSI by the decision tree method. Since the MSI contained a small amount of spectral information (the MODIS image had 21 bands and the Landsat 8 image had 7 bands), if the spectral genes were extracted from MSI by the LIS strategy, most of the spectral DNA code words would be eliminated. It was hard to correctly identify the oil slicks by the remaining spectral genes. As a result, many pixels of oil spills and seawater would be wrongly determined. However, if the spectral genes were extracted from MSI by the LID strategy, several unique spectral genes could be extracted. The decision tree method could make the most of their uniqueness to distinguish between each of them rapidly. Thus, the SGE method with the LID strategy performed excellently for detecting oil spills in MSI (Figures 8 and 9). Experimental results indicated that the SGE method strengthened the oil spill detection ability. In addition, it was discovered that the SGE method with the LIS strategy was more suitable for detecting oil spills in HSI. Its best elimination rates were

0.8~1.0. The SGE method with the LID strategy was more suitable for detecting oil spills in MSI. Its best elimination rates were 0.5~0.7.

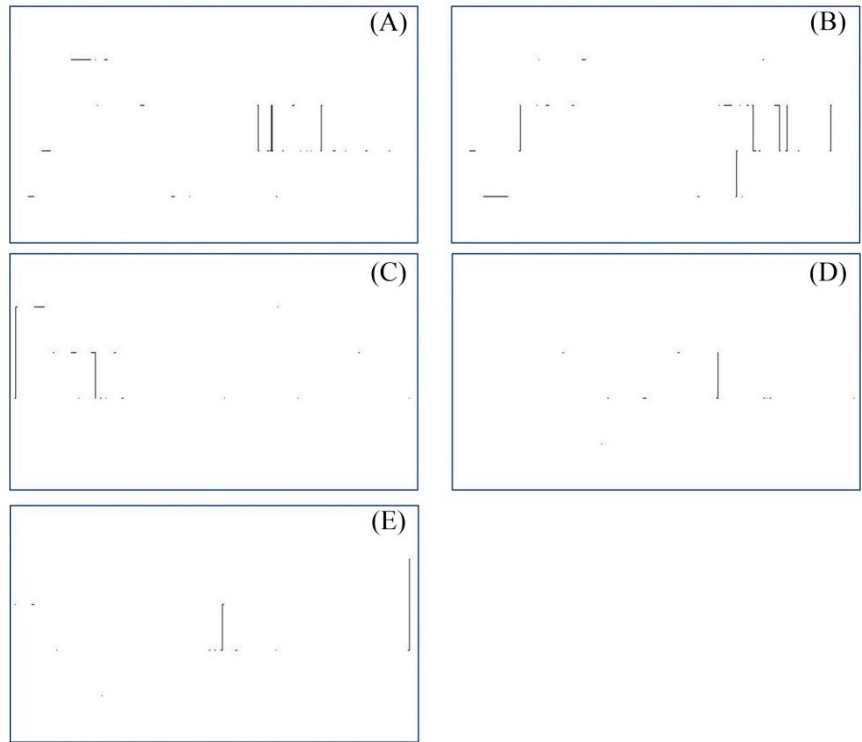

**Figure 12.** The spectral genes extracted by the LID strategy from the experimental AVIRIS image. (**A**) the spectral genes of emulsions, (**B**) the spectral genes of oil slick of code 5, (**C**) the spectral genes of oil slick of code 4, (**D**) the spectral genes of sheens, and (**E**) the spectral genes of seawater.

### 4.2. Superiority of the SGE Method

Compared to the traditional spectral indices and classification methods, the SGE method had a specific superiority. Figure 13 shows the mapping results of the oil slick thickness in the experimental AVIRIS image through the near infrared spectral absorption index based on the conservative strategy (B) and aggressive strategy (C), respectively [8]. In the results, the thickness of oil slicks was stretched from 2 mm (bright) to 0 mm (dark). However, only the thick oil slicks were identified by the near infrared spectral index because thick oil slicks had higher heat capacity. They could express high infrared reflectance and show clear absorption characteristics. Thin oil slicks could not reserve heat. As a result, they could not be detected by the near infrared spectral absorption index. Quantitative inversion methods based on field research were beneficial for oil spill assessment, but they usually cost weeks to produce the results. Figure 13D shows the result produced by the SGE method. In this result, red, black, yellow, and cyan pixels represent the identified emulsions, oil slicks of code 5, oil slicks of code 4, and sheens, respectively. It could be observed that the result produced by the SGE method was more consistent with the oil spill distribution in Figure 13A. In addition, the SGE method could produce the oil spill detection results rapidly with a small set of samples.

The performance of the SGE method was compared with several other classical classification methods. Figures 14–16 show the oil spill detection results by minimum distance (MD), spectral angle mapper (SAM), support vector machine (SVM), artificial neural network (ANN), and the SGE method on the experimental AVIRIS images [36,37]. All the results were produced with the same samples (Table 1).

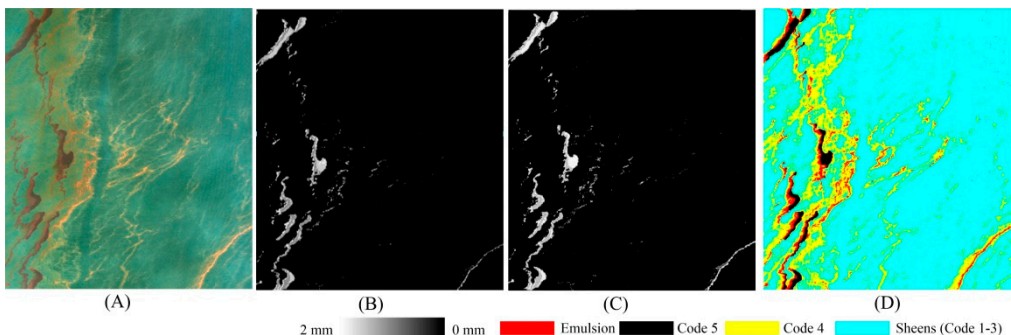

**Figure 13.** Detection results produced by the spectral near infrared index (**B**,**C**) and the SGE method (**D**) on the experimental AVIRIS image 1 (**A**).

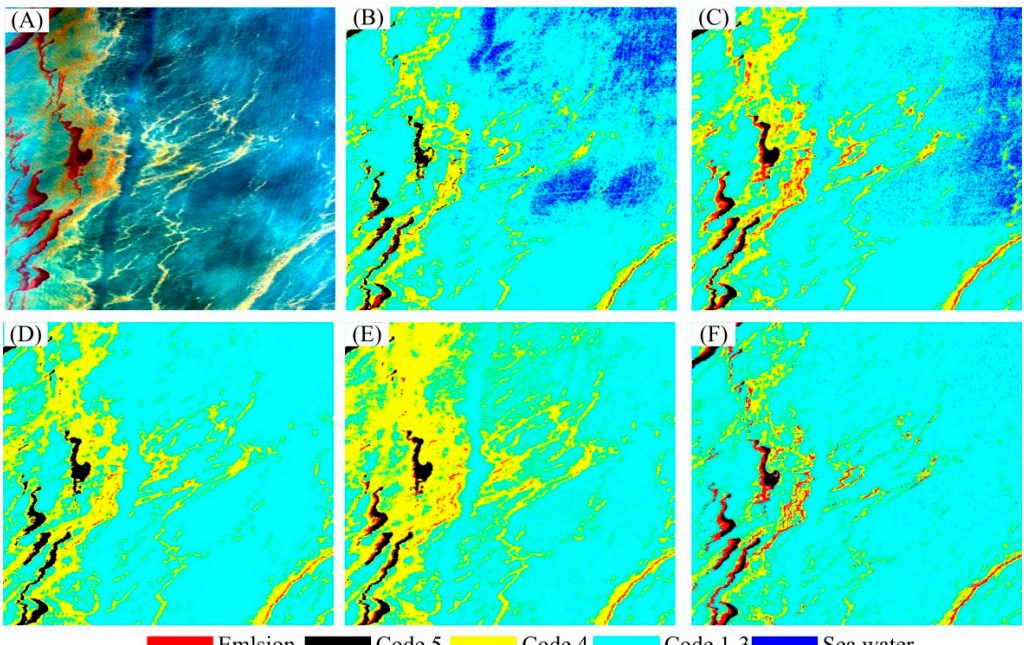

**Figure 14.** Detection results produced by MD (**B**), SAM (**C**), SVM (**D**), ANN (**E**), and the SGE method (**F**) on the experimental AVIRIS image (subarea-1) (**A**).

From the results on the experimental AVIRIS images, it could be found that all of the methods obtained nice accuracies of oil slicks of code 5 because they had stable compositions and particular spectral reflectance (Figure 14B). The MD method could identify oil slicks of code 4 well (Figures 14B and 15B). However, it could only extract a small part of emulsions from the AVIRIS images because the spectra of emulsions varied drastically, and the overall spectral distance could not distinguish emulsions from the other oil slicks. In addition, MD could not distinguish sheens from the unpolluted seawater well. SAM showed excellent performance of detecting oil slicks of code 5 and emulsions (Figures 14C and 15C). However, a large number of sheens were misidentified in the results produced by SAM. Compared to the traditional spectral matching method, SVM and ANN had obvious advantages in detecting sheens (Figure 14D,E). However, SVM and ANN could hardly extract emulsions around oil slicks of code 5. ANN obtained a more aggressive result of oil slicks of code 4 than SVM. The SGE method obtained a more conservative result of oil slicks of code 4 than ANN and SVM (Figure 14F). However, the SGE method could detect oil slicks of code 5 and emulsions accurately. In addition, MD, SAM, and ANN could not extract the sheens around the oil spill boundary (Figure 16). SVM could only detect a part of the sheens around the boundary. However, the SGE method could accurately detect the sheens around the oil spill boundary. The performance of ANN and

SVM was restricted because of the limited training samples. However, the SGE method could produce satisfying oil spill detection results using a small set of samples.

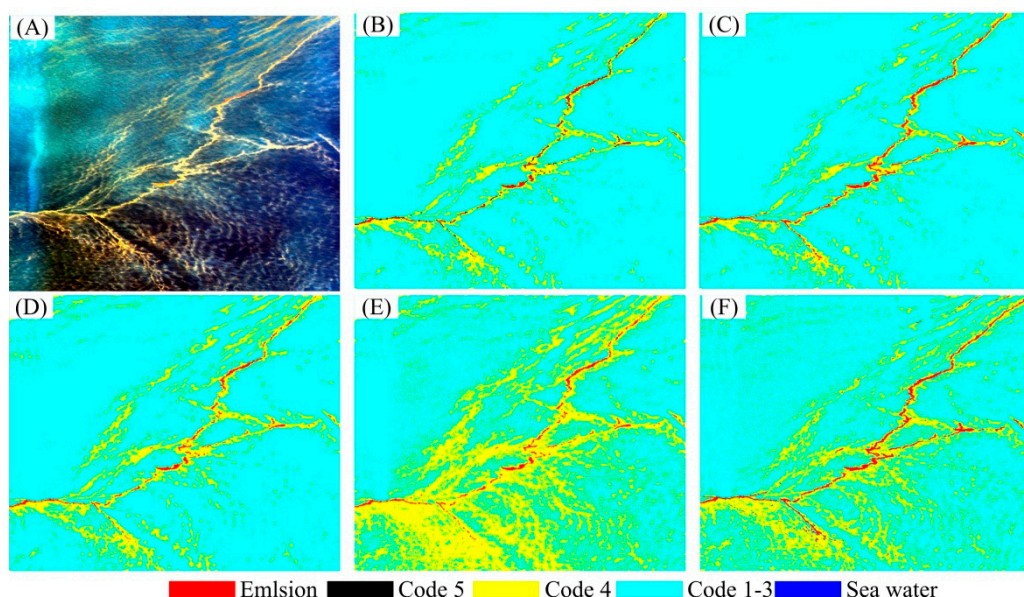

**Figure 15.** Detection results produced by MD (**B**), SAM (**C**), SVM (**D**), ANN (**E**), and the SGE method (**F**) on the experimental AVIRIS image (subarea-2) (**A**).

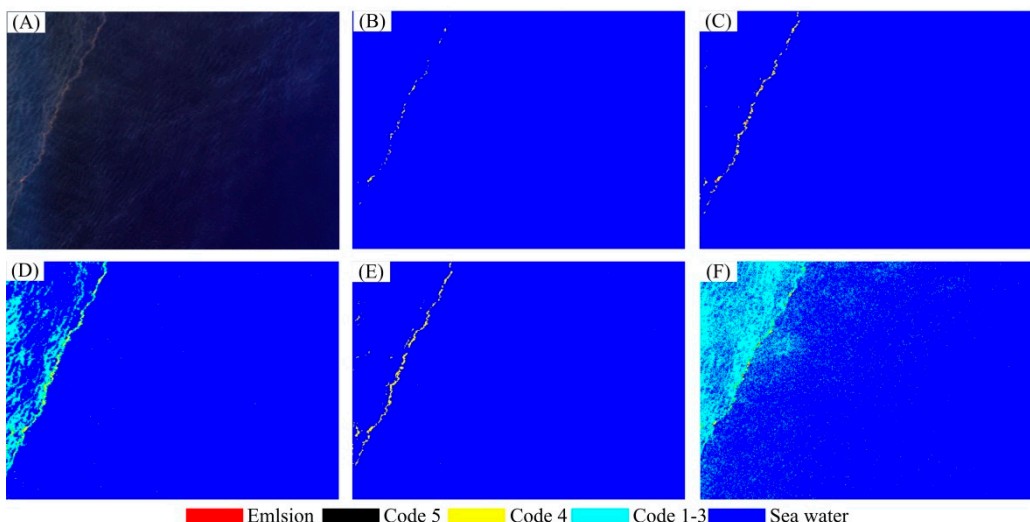

**Figure 16.** Detection results produced by MD (**B**), SAM (**C**), SVM (**D**), ANN (**E**), and the SGE method (**F**) on the experimental AVIRIS image (subarea-3) (**A**).

## 5. Conclusions

Oil spill incidents are harmful to the marine ecological environment. For the sustainability of our world development, it is important to monitor marine oil spills. Since hyperspectral and multispectral remote sensing technologies could be utilized to detect marine oil spills on a large scale, we focused on establishing an accurate and applicative method to detect oil spills in HSI and MSI. Illuminated by the distinctive description ability of the spectral DNA encoding method, the SGE method based on the population genetic characteristics was proposed to detect the oil spills in HSI and MSI. A parameter of the elimination rate and two spectral gene extraction strategies, the LIS strategy and LID strategy, were proposed to mine the spectral genes of oil slicks. Different hyperspectral datasets and multispectral datasets were used to validate the performance of the SGE method. The

results indicated that (1) the performance of the proposed method was excellent. (2) Oil spills in different HSI and MSI could be mapped correctly by the proposed method. In addition, (3) the SGE method of the LIS strategy was more suitable for detecting oil spills in HSI when the elimination rate was 0.8~1.0. The SGE method of the LID strategy was more suitable for detecting oil spills in MSI when the elimination rate was 0.5~0.7. As for future studies, it was suggested to combine the SGE method with the backpropagation (BP) neural network. The encoded code words could be considered attributes of samples. Using the code words encoded by different scales to construct BP neural networks would be convenient and effective. In addition, other deep learning methods might be able to use the SGE result to detect ground objects in HSI and MSI.

**Author Contributions:** Conceptualization, D.Z.; methodology, D.Z. and B.T.; software, D.Z. and B.T.; validation, H.Z., R.D. and B.T.; formal analysis, D.Z.; investigation, D.Z.; resources, B.T.; data curation, H.Z.; writing—original draft preparation, D.Z.; writing—review and editing, R.D.; visualization, H.Z.; supervision, H.Z.; project administration, D.Z.; funding acquisition, D.Z., B.T., H.Z., and R.D. All authors have read and agreed to the published version of the manuscript.

**Funding:** This research was funded by the Open Fund of Key Laboratory of Exploration Technologies for Oil and Gas Resources (Yangtze University), Ministry of Education, grant number K2021-6. This research was funded by the Science and Technology Program of Guangdong Province, China, grant number 2021B1111610001 and 2021B1212100003. This research was funded by the Key Scientific Research Projects of Colleges and Universities in Henan Province, grand number 20B420001.

**Institutional Review Board Statement:** Not applicable.

**Informed Consent Statement:** Not applicable.

**Data Availability Statement:** Not applicable.

**Conflicts of Interest:** The authors declare no conflict of interest. The funders had no role in the design of the study; in the collection, analyses, or interpretation of data; in the writing of the manuscript; or in the decision to publish the results.

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
