# Peer review of "Monitoring Marine Oil Spills in Hyperspectral and Multispectral Remote Sensing Data by the Spectral Gene Extraction (SGE) Method"

_sustainability, doi:10.3390/su142013696_

Round 1
Reviewer 1 Report
This work presents a spectral gene extraction method for monitoring marine oil spills.
The manuscript needs to go through a minor revision before it can be considered for publication in sustainability.
1. In the first paragraph, it is suggested to provide some oil spill incidents to explain its harm and hazard.
2. The figure resolution needs to be increased for most figures.
3. The scale label is suggested added in Figure 8 and 9, like Figure 5.
4. The conclusion is not sufficient, more contents are suggested to supply.
Author Response
Thanks for your advice for our manuscript. The responses and revisions are listed according to your advices.
- In the first paragraph, it is suggested to provide some oil spill incidents to explain its harm and hazard.
We are sorry for the missing of the explanation of oil spill incidents. It is important to explain the harm of oil spill incidents by examples. Thus, we added the examples of Gulf of Mexico oil spill in 2010 and the Japanese ship (WAKASHIO) oil spill in 2020 in the first paragraph (marked yellow). The harm of oil spills is explained. We hope it was a suitable revision.
- The figure resolution needs to be increased for most figures.
We are sorry for the low resolution figures. The resolutions of the old figures were the screen resolution. Thus they seem blurry in the manuscript. Following your advice, we have increased the resolutions of the figures to 300*300 PPI. Hope the revised figures were satisfying.
- The scale label is suggested added in Figure 8 and 9, like Figure 5.
Thanks for your advice. It is our negligence to exhibit remote sensing images without scale labels and compasses. The scale labels and the compasses were added to Figure 8 and Figure 9 by ArcGIS 10.2. Hope it was suitable.
- The conclusion is not sufficient, more contents are suggested to supply.
We are sorry for the insufficient conclusion. The key processes of the proposed method and the detailed experimental conclusions are missing in this part. Thus, we overwrite the part of conclusions (marked yellow). The research significance, technology, method details, and experimental conclusions were provided in the revised part of conclusions, respectively. We hope the revised version was sufficient.
15 October 2022
Dong Zhao

Reviewer 2 Report
Some English grammatical errors were found. Consider English proofread.
Abstract does not have flow. Consider restructure to make it a clear view of what you are intending to present.
Abstract dont have a conclusion, so what are the significance from ur findings?
Line 37: Restructure the sentence for a scientific purpose.
Line 87-94: U dont present results here. State clearly what are ur objectives?
Line 99: Dont start ur paragraph with figure. Restructure.
Line 141: Is the formula from Alloy and XU? What does Nb means'? State clearly.
232: Reference?
239: More details on this flight
246: Label what ocean is this, where is new orleans.
249: Software update please
272-300: Lack of reference, please add in citation
334-336: Discuss more on this
352: Discuss more with past studies
Overall discussion, there are no related past studies to support ur findings. Consider add more.
Line 429: No need to conclude this as this was not part of your experiment.
Line 438: Recommendations for future study?
Author Response
Thanks for your advice for our manuscript. The responses and revisions are listed according to your advices.
Some English grammatical errors were found. Consider English proofread.
We are sorry for the grammatical errors. According to your advice, the manuscript was checked and revised by a colleague who got a PhD from Texas A&M University. After that, we checked the manuscript again. Hope the revised version was satisfying. We just have 5 days to revise the manuscript. If there are still many grammatical errors, we would like to revise it further.
Abstract does not have flow. Consider restructure to make it a clear view of what you are intending to present.
Thanks for your advice. We discussed about the abstract and agreed with you. The old version of abstract was not clear. Now, we restructured it. The following was the context of the revised abstract. It contained four parts: 1) why to study oil spill detection methods in HSI and MSI. 2) The shortcomings of the traditional oil spill methods. 3) The main parts of the proposed method (the spectral DNA encoding method, the LIS and LID strategy, and the elimination parameter). 4) Experimental results and the conclusions. We hope the flow could make the abstract clear.
â‘ Oil spill incidents threat the marine ecological environment. Detecting sea surface oil slicks by remote sensing images provides support for efficient treatment for oil spills. It is important for the marine sustainable development. â‘¡ However, traditional methods based on field analysis are time-consuming. Spectral indices lack applicability. In addition, traditional machine learning methods strictly rely on the training and testing samples which were in short supply in oil spill images. â‘¢ Inspired by the spectral DNA encoding method, a spectral gene extraction (SGE) method was proposed to detect oil spills in hyperspectral images (HSI) and multispectral images (MSI). The SGE method contained a parameter and two strategies. The parameter of elimination was based on the population genetic frequency. It was used to control the number of spectral gene. The spectral gene extraction strategies, named largest in-class similarity (LIS) strategy and largest inter-class difference (LID) strategy, were proposed to mine the spectral genes by oil spill samples. The oil spills would be determined by calculating the similarity of the extracted spectral genes and the DNA encoded images. â‘£ In this research, the SGE method was validated by two AVIRIS images of the Gulf of Mexico oil spill, one MODIS image of the Gulf of Mexico oil spill, and one Landsat 8 image of a Persian Gulf oil spill. The oil spills in different remote sensing images could be detected accurately by the proposed method in small set of samples. Experimental results indicated that the proposed method was suitable for detecting marine oil spills in AVIRIS, MODIS, and Landsat 8 images. In addition, the SGE method with LIS strategy was more suitable for detecting oil spills in HSI. The proper elimination rate of it was 0.8~1.0. The SGE method with LID strategy was more suitable for detecting oil spills in MSI. The proper elimination rate of it was 0.5~0.7.
Abstract dont have a conclusion, so what are the significance from ur findings?
Sorry for the bad structure of the old abstract. It was fuzzy of the conclusion in the old version of abstract. We restructured it and pointed the conclusion of the research: the proposed method was suitable for detecting marine oil spills in AVIRIS, MODIS, and Landsat 8 images. In addition, the SGE method with LIS strategy was more suitable for detecting oil spills in HSI. The proper elimination rate of it was 0.8~1.0. The SGE method with LID strategy was more suitable for detecting oil spills in MSI. The proper elimination rate of it was 0.5~0.7. Hope it was a satisfying conclusion.
Line 37: Restructure the sentence for a scientific purpose.
Sorry for the bad structure of the sentence. We revised it to “Oil spill incidents usually cause serious marine ecological disasters. They make crude oil or fuel oil leak into the marine environment.”.
Line 87-94: U dont present results here. State clearly what are ur objectives?
We are sorry to list the conclusions without presenting experimental results. We revised this part and simply stated the experimental results here: “The experimental results showed that the SGE method could correctly detect the oil spills in different HIS and MSI.”. In addition, according to the other reviewer’s advice, the detailed conclusions were removed from the introduction part to the conclusion part. Thus, this part was revised to “In this research, the proposed method was validated through two AVIRIS images obtained during the Gulf of Mexico oil spill incident in 2010, one MODIS image of Gulf of Mexico oil spill incident in 2010, and one Landsat 8 image of Persian Gulf in 2013. The experimental results showed that the SGE method could correctly detect the oil spills in different HIS and MSI. The description of the proposed method, experimental results, discussions, and conclusions were provided below.” We hope it was a suitable revision.
Line 99: Dont start ur paragraph with figure. Restructure.
Thanks for your advice. The figure was moved into the test body.
Line 141: Is the formula from Alloy and XU? What does Nb means'? State clearly.
The formula 1-9 was from Hongzan Jiao (reference 15). We revised the second of section 2.2 and mentioned the provenance of the formula. In the formulas, Nb is the spectral band number of samples and pixels. It was revised and stated at the head of the third paragraph of section 2.2. We hope it was a suitable revision.
232: Reference?
Sorry, Line 232 was about the work flow of the proposed method. You mean the description of the experimental data needs reference? The situation of GoM oil spill could refer to reference 19 and 20. We added it in the section 3.1.
239: More details on this flight
Thanks for your advice. We searched the flight details in the JPL data portal. The details of the AVIRIS data were added at the section 3.1.
246: Label what ocean is this, where is new orleans.
Thanks, we followed your advice and labeled the New Orleans by a yellow star. The ocean was labeled as Gulf of Mexico.
249: Software update please
Following your advice, the software was updated to ENVI 5.5.
272-300: Lack of reference, please add in citation
Thanks for your advice. It is our mistake missing the citations when describing the oil spill physical characteristics although the references have been cited in the other sections. We had added the citations in these two paragraphs.
334-336: Discuss more on this.
Thanks for your advice. We are sorry for ignoring to compare and analyze the result in detail before providing the result description. We revised this part and added more description of the discussion experimental results. “Since the SGE method could filter the redundant spectral DNA code words by the LIS and LID strategies, and the key spectral genes were reserved to detect oil slicks. The expression ability for oil slicks was strengthened by the spectral genes. However, the original spectral DNA chains could not distinguish the oil slicks with similar spectral signals, such the sheens and oil slicks of code 4, because the redundant code words increased their similarity. Thus, the original spectral DNA chains tended to produce many misidentified pixels of oil slicks. It could be clearly observed that detection result produced by the spectral genes was much more accurate than the result produced by the original spectral DNA chains.”. We hope it was suitable to support the discussion.
352: Discuss more with past studies. Overall discussion, there are no related past studies to support ur findings. Consider add more.
Thanks for your advice. The section 4.1 was about discussing the proposed two spectral gene extraction strategies. In the first paragraph, we compared them (Figure 11, 12) with the spectral DNA encoding method (Figure 2). In this paragraph we analyzed why the SGE method with the LID and LIS strategy performed better than the original spectral DNA chains. Then, in the second paragraph, we revealed the advantages of the LIS strategy (the LIS strategy was more suitable for the HSI). In the third paragraph, we revealed the advantages of the LID strategy (the LID strategy was more suitable for the MSI). Since the proposed method was based on the spectral DNA encoding method, the spectral DNA encoding method was the only related past studies discussed in the section. However, we compared the proposed method with other traditional methods in section 4.2. Hope it was suitable.
Line 429: No need to conclude this as this was not part of your experiment.
Yes, you are right. All the experiments are about detecting oil spill caused by incidents which would leak a large amount of oils. Natural oil spills were not concluded in this research. We did some experiments on the natural oil spill detection. It was a mistake to add this at the conclusion. We are sorry about that and we have deleted the sentence. Thanks again.
Line 438: Recommendations for future study?
Following your advice, the recommendations for the future study were provided at the end of the conclusion part. “As for the future study, it was suggested to combine the SGE method with the back propagation (BP) neural network. The encoded code words could be considered as attributes of samples. Using the code words encoded by different scales to construct BP neural network would be convenient and effective. In addition, other deep learning methods might be possible to use the SGE result to detect ground objects in HSI and MSI.” In fact, we are doing researches on these aspects.
15 October 2022
Dong Zhao

Reviewer 3 Report
This study is very interesting. Authors proposed one spectral gene extraction method to extract oil slicks. Compared with exising studies, authors found that their methods could detect marine oil spills accurately using small samples. Generally, this is a good manuscript with good novelty. After minor modification, this manuscript could be published in Sustainaibility journal.
Line 89-94: this part is the key findings of your work. It is inappropriate to put it in introduction part, it should be in abstract, result or conclusion, not in introduction. Please modify.
Line 99: improve the figures' dpi in your manuscript.
Line 429-438: in conclusion part, you should provide the key results you described in abstract part. Existing descrition is too vague.
Author Response
Thanks for your advice for our manuscript. The responses and revisions are listed according to your advices.
Line 89-94: this part is the key findings of your work. It is inappropriate to put it in introduction part, it should be in abstract, result or conclusion, not in introduction. Please modify.
Thanks for your advice. It is indeed inappropriate to put the conclusions in the introduction part. It makes the manuscript wordy and redundant. We deleted the conclusions in the introduction. Since the brief description of the conclusions was provided at the end of the abstract, it was removed to the conclusion part. We hope it was a suitable revision.
Line 99: improve the figures' dpi in your manuscript.
We are sorry for the low resolution figures. The resolutions of the old figures were the screen resolution. Thus they seem blurry in the manuscript. Following your advice, we have increased the resolutions of the figures to 300*300 PPI. Hope the revised figures were satisfying.
Line 429-438: in conclusion part, you should provide the key results you described in abstract part. Existing descrition is too vague.
We are sorry for the vague conclusion. The key processes of the proposed method and the detailed experimental conclusions are missing in this part. Thus, we overwrite the part of conclusions. The research significance, technology, method details, and experimental conclusions were provided in the revised version of conclusions, respectively. In addition, the key findings in the introduction part were described in the conclusion part. It made the conclusion part more substantial. We hope the revised version was sufficient.
15 October 2022
Dong Zhao
